# Feeding and Nutritional Factors That Affect Somatic Cell Counts in Milk of Sheep and Goats

**DOI:** 10.3390/vetsci10070454

**Published:** 2023-07-11

**Authors:** Anna Nudda, Silvia Carta, Gianni Battacone, Giuseppe Pulina

**Affiliations:** Department of Agricultural Sciences, University of Sassari, Viale Italia, 39, 07100 Sassari, Italy; scarta2@uniss.it (S.C.); battacon@uniss.it (G.B.); gpulina@uniss.it (G.P.)

**Keywords:** somatic cell count, SCC, sheep and goat nutrition, minerals and vitamins, vegetable oils

## Abstract

**Simple Summary:**

Milk somatic cells are a sign of udder discomfort and constitute a problem in the process of making cheese from sheep and goats’ milk. Their control, which can be carried out through veterinary means, is also possible through food by administering mineral and vitamin supplements to the animals or by using substances with anti-inflammatory effects, such as polyphenols, or antimicrobial effects, such as essential oils.

**Abstract:**

The purpose of this quantitative review is to highlight the effects of feeding strategies using some mineral, vitamin, marine oil, and vegetable essential oil supplements and some agri-food by-products to reduce SCCs in the milk of sheep and goats. According to the results, only specific dietary factors at specific doses could reduce SCCs in the milk of dairy sheep and goats. The combination of Se and vitamin E in the diet was more effective in sheep than in goats, while the inclusion of polyphenols, which are also present in food matrices such as agro-industrial by-products, led to better results. Some essential oils can be conveniently used to modulate SCCs, although they can precipitate an off-flavoring problem. This work shows that SCCs are complex and cannot be determined using a single experimental factor, as intramammary inflammation, which is the main source of SC in milk, can manifest in a subclinical form without clinical signs. However, attention to mineral and vitamin supplementation, even in the most difficult cases, such as those of grazing animals, and the use of anti-inflammatory substances directly or through by-products, can improve the nutritional condition of animals and reduce their SCCs, offering undeniable benefits for the milk-processing sector as well.

## 1. Introduction

Milk somatic cells (SC) include two categories of cells: immune system cells derived from the blood, which constitute more than 95% of a total somatic cell count (SCC), and epithelial cells derived from the desquamation of the mucosa lining the inside of the udder, which constitute the remaining 0–3% of the SCC. SCC from blood comprises polymorphonuclear neutrophils (PMNs), macrophages, and lymphocytes [1,2]. In the count of somatic cells in milk, epithelial cells and cytoplasmic residues, resulting from apocrine secretion, are also present; these cells and residues are shed into milk from the apical portion of mammary secretory cells [3]. SC play a key role in the defense of the mammary gland against environmental pathogens and are an important indicator of udder health. Additionally, in raw milk, they reflect the general health status of an animal [4,5]. The number and distribution of various cell types are influenced by physiological and pathological conditions. Differences between species have also been reported [6]. Milk SCCs are usually higher in goats than in sheep, and macrophages are the major cell type present in the milk (45–88%) of uninfected sheep, whereas PMNs comprise the major cell type in the milk from healthy, uninfected goats (4–74%). An increase in the milk SCC in sheep milk has been associated with a decrease in milk yield and a change in milk composition along with a decrease in casein and lactose content, while whey protein and its fractions (immunoglobulin, serum albumin, and lactoferrin) increase [7]. These changes in milk composition mostly derive from paracellular passage through disrupted tight junctions during intramammary inflammation and infection [8]. Intramammary infection is the major cause of an increased SCC in milk. However, SCCs are also increased through various physiological factors such as the stage of lactation, parity, fertility, and milking practices [9]. Physical stress, e.g., stress during transport, is also considered an additional factor that increases SCCs.

The immune defenses of sheep and goats can also be influenced by specific nutrients supplemented during the production and reproduction phase. Dairy animals should regularly be fed minerals and vitamins in addition to energy- and protein-balanced diets.

Nutritional errors (such as those regarding formulation, composition, and the administration of rations) can affect the metabolism of lactating animals, predisposing the mammary gland to inflammation and thus increasing the probability of the occurrence of mastitis, which, importantly, is the major factor influencing milk SCCs in dairy ruminants, including sheep and goats. Among the most common feeding errors, energy deficiencies [10,11,12], unbalanced energy/protein ratios [13], protein deficiencies and/or excess NPN in the diet [13], and improper diets that predispose animals to metabolic disorders such as subacute acidosis or ketosis [14] can affect milk SCCs. A detailed analysis of the relationship between feeding and SCCs was reported in one of our previous reviews, which is referenced herein for further information [15]. Antioxidative agents, such as vitamins and minerals, can protect cells from free radicals and prevent and delay cell damage. The supplementation of dairy animals with high SCC with vitamins, such as vitamins A and E, and minerals, such as selenium, zinc, and copper, was found to be effective in reducing their SCCs and ensuring early recovery from mastitis [16,17]. Despite the large number of studies evaluating the beneficial effects of various supplements with antioxidant properties on animal health status and milk production traits, there is a lack of research on the quantitative effects of feeding and nutritional factors on milk SCCs. Deeper knowledge of the relationship between diet and SCC may help in the management of supplements with respect to reducing susceptibility to disease among dairy animals.

Moreover, an increase in SCCs has negative implications for the dairy industry, i.e., a reduction in milk yield and the worsening of milk quality. In fact, the high content of SC in milk is related to a decrease in lactose and casein content and to an increase in serum protein, sodium, chloride, pH, and proteolytic activity [18,19]. These factors lead to the worsening of milk coagulation properties, particularly an increase in coagulation time, and a decrease in curd firmness and cheese yield [20,21]. The strong influence of SCCs on the dairy industry emphasizes the increasing need to use all management and nutritional strategies to reduce intramammary infections. The use of some minerals, byproducts, or oils in the diets of small ruminants could improve animal health and, consequently, milk quality due to the presence of important compounds in these products. However, the results in this field are not unique because they depend on several factors, such as the species, the doses used in the conducted experiment, and the methods of trial implementation.

The aim of this review is to quantify the efficacy of supplementing the diets of lactating sheep and goats with antioxidant minerals and vitamins, as well as marine oils and plant essential oils, or specific agri-food by-products, in reducing SCC in their milk.

## 2. Materials and Methods

In 2023, a systematic literature search was carried out on Google Scholar, Web of Science, and PubMed using the following keywords: by-products, polyphenols, energy balance, minerals, vitamins, essential oils, marine oil, vegetable oil, fat supplements and milk yield, milk composition, and SCCs. This research was repeated by including the terms sheep, goat, cow, and ruminant as well as cattle, sheep, and goats. The inclusion criteria were milk yield, its main components (fat and protein), and the SCC of milk. The initial search yielded 58 articles, including 29 involving sheep and 19 involving goats. Some feeding experiments carried out by our research groups on small ruminants were included in the database, where SCCs were routinely measured. The dataset was prepared by including the results of the control and experimental groups; the degree of variation of each experimental group from the control group was calculated and expressed as a percentage of the log of SCCs, and data originally expressed as SCC were log-transformed. Statistically significant differences (*p* < 0.05), as declared by the authors, are shown in bold in the tables. Otherwise, the percentage differences are not significant or were not reported by the authors.

## 3. Results and Discussion

The review gives an overview (summarized in tables) of the main effects of nutritional factors that could affect SCCs when measured in sheep and goats.

The effects of minerals, vitamins, byproducts, marine and vegetable oils, and feed strategies on SCCs—expressed as the extent of the variation of each experimental group from the control group—are shown in Table 1, Table 2, Table 3 and Table 4. Mineral supplementation led to an overall decrease in SCCs, emphasized by the combined effect of vitamins, in both sheep and goats (Table 1). The effects of byproducts on milk SCCs were extremely variable (Table 2). This result was expected, as these byproducts are included in small ruminants’ diets as a convenient solution for the valorization of the residues generated through agricultural activities. However, most of them contain considerable amounts of bioactive compounds, including polyphenols, which can have potential nutraceutical effects on animals, if used in the correct dose. The non-univocal results could depend on the type of byproduct, the type of polyphenol, and the dosage applied in the animal’s diet. Marine and vegetable oils showed interesting results regarding the reduction in the SCCs in both sheep and goat milk (Table 3). The supplementation of oils to lactating ruminants has increased due to certain oils’ antimicrobial properties. Essential oils can inhibit the growth of some bacteria by interacting with microbial cell membranes. However, the effects of some of these essential oils are diet-dependent, and their use may only be beneficial under certain conditions and production systems.

### 3.1. Effects of Mineral and Vitamin Supplementation

Specific minerals and trace elements are strongly associated with the health of the mammary gland and, therefore, with the SCC in the corresponding milk. In fact, minerals are involved in several physiological processes because they are components of antioxidant enzymes, such as Se in glutathione peroxidase (GPx) and Zn, Mn, and vitamin E in superoxide dismutase (SOD). These enzymes are essential for maintaining cellular health due to the protective role they play against oxygen radicals and free radicals [22]. To ensure optimal immunity among dairy animals, zinc is an essential micronutrient. Almost all immune cells, including neutrophils, macrophages, natural killer cells, T-cells, and B-cells, are zinc-dependent, so zinc acts as a gatekeeper for immune function [23].

Specifically, SOD is a major antioxidant enzyme in cells that participates in a dismutation reaction to convert superoxide into the less-toxic substances of hydrogen peroxide and dioxygen. In cells, GPx is the most important enzyme with respect to scavenging hydrogen peroxide, in which it converts this molecule into water. SOD and GPx can directly counteract oxidant attack and protect cells from DNA damage. For this reason, appropriate mineral and vitamin supplementation in the diets of sheep and goats has been shown to have an inhibitory effect on milk SCC (Table 1). Morgante et al. [24] demonstrated a −10% reduction in SCC in dairy sheep supplemented with Se and vitamin E. They also found a negative correlation between SCCs and GPx activity and a reduction in the percentage of polymorphonuclear neutrophils (PMN) during the first 90 days of lactation. The synergistic effect between vitamin E and Se had a positive effect on mammary gland health and, consequently, on SCCs. Koutsoumpas et al. [25] showed that sheep given 3500 IU of vitamin A kg^−1^ body weight via intramuscular injection every 3 months reduced milk SCCs (−24.54%) at the 12th week of lactation, indicating that there is an increased risk of mastitis when vitamin A is deficient in the diet. The protective role of vitamin A in sheep was also confirmed by Raynal-Ljutovac et al. [26], who showed a reduction in milk SCCs in sheep given an intramuscular administration of 200 mg of beta-carotene, a precursor of vitamin A. Carotenoids are important free radical scavengers, particularly with respect to their role as effective quenchers of singlet oxygen, and can prevent the subsequent formation of secondary reactive oxygen species (ROS) [27]. Vitamin A plays an important role in epithelial integrity and stability, and a deficiency in this vitamin can predispose animals to infection by pathogens [25]. However, the effective influence of vitamin A or β-carotene on the incidence of mastitis has yet to be fully demonstrated. Zn administration did not show significant effects on the SCCs in sheep milk [28,29], but it had a significant positive effect on goat SCCs. For example, the administration of 1 g/d Zn-Met to lactating goats reduced SCCs by 5.15% compared to the employed control group [30].

Supplementation of goat diets with 20 ppm of inorganic Zn and with 10 and 20 ppm of nano-Zn led to a reduction in SCCs; the greatest reduction was found when Zn was administered as nano-Zn (−9.03% at 20 ppm). The better efficacy of nano-Zn than inorganic Zn in reducing SCCs was probably due to the better availability and efficient utilization of the former. The positive influence of Zn on udder health has been reported by several studies on dairy cows [31,32,33]. Indeed, Zn plays an important role in maintaining normal epithelial barrier integrity and the structural integrity of the mammary gland [34], as reported above. Zn is also involved in the formation of keratin, which is important for protecting the teat canal from infection and preventing new infections [30].

Similar to dairy sheep, the supplementation of Se alongside vitamin E reduced SCCs in goat milk (−31.79%) [35]. Se is an important element that could have a positive effect on mammary health due to its involvement in the antioxidant system, which improves the defense of mammary cells. The supplementation of a goat diet with 0.23 μg of Se resulted in a −14.93% reduction in SCCs. It is important to point out that the reduction in the SCC for each feeding treatment refers to the log-transformed data. The magnitude of the variation when using the raw CCS data is much larger than the values when expressed on a logarithmic scale. For example, the magnitude of variation reported by Morgante [24] is 10% in logarithmic form and 53.5% in CCS form.

The supplementation of minerals and vitamins could be an interesting strategy with which to improve mammary health and reduce SCCs in milk. Furthermore, the fact that the strongest positive effect on SCCs was observed when minerals and vitamins were used in combination suggests that they have a synergic effect.

**Table 1 vetsci-10-00454-t001:** Effects of supplementing ewe and goat rations with minerals and vitamins on milk somatic cell counts. Data are reported as proportional differences between the treatment group, at the respective supplementation level, and the control group (bold differences are considered significant, with *p* < 0.05).

Species	Dietary Treatment	mg/kg of DM	Log SCC	Milk Yield	Lactose	References
Dairy sheep	Zn500	500	0.53%	-	-	[29]
	Zn1000	1000	−0.53%	-	-	
	Zn	113	−1.75%	-	-	[28]
	Se + Vit E	5.1	**−10.00%**	-	-	[24]
	Se *	2.9	−1.37%	-	-	[36]
	Vit A *	- **	**−24.53%**	-	-	[25]
Dairy goat	Inorganic Zn *	20	**−4.06%**	3.30%	−5.63%	[37]
	Nano Zn *	10	**−5.37%**	7.69%	−4.63%	
	Nano Zn *	20	**−9.03%**	7.69%	−1.21%	
	Zn methionine *	5000 ***	**−5.15%**	2.50%	-	[30]
	Se + Vit E	0.14 Se11 VitE ***	**−31.79%**	**7.14%**	3.87%	[35]
	Se *	-	**14.93%**	-	-	[38]
	Organic acids and pure botanic	10 ****	10.1%	1.6%	0.5%	[39]

* Data originally expressed as SCCs were log-transformed by authors. ** In this experiment, the animals received a dose of Vitamin A every 3 months via intramuscular injection. *** Dose expressed as mg/of DMI. **** Expressed as g/head. Bold values indicate significant differences (*p* < 0.05) compared to the control group as reported in the original paper.

### 3.2. Effects of Polyphenols and Products Containing Polyphenols

The incorporation of by-products in the diets of small ruminants is a common practice used to recycle products from the agro-industrial chain, thus reducing the cost of animal feed. The use of some by-products has a positive effect on animal performance, milk composition, and milk quality [40,41]. However, less is known about the effects of these by-products on animal health and, consequently, SCCs. In fact, several studies that investigated the effect of by-products on animal performance and milk quality have not reported the effect of the same by-products on SCCs. Due to the presence of bioactive compounds in agro-industrial by-products, a reduction in the SCCs of milk from animals fed with these products could be expected. As shown in Table 2, the effects of different by-products on SCCs were not clear and depended on the type of by-products and the doses used in the experiments. In sheep milk, the authors of [42] reported a significant reduction in SCC equal to 12.66% when considering the inclusion of 100 g/d of cocoa hulls in the diet. The beneficial effect of cocoa hulls was not supported in [43], whose authors only found a numerical reduction in SCC. In sheep blood, the ingestion of cocoa by-products led to a reduction in basophils [42], which play a critical role in immunity against parasites [44]. Cocoa by-products containing theobromine [42,45] (252 mg/kg of DM), a bitter alkaloid, have been reported to exert anti-inflammatory and antioxidant actions [46] if the compound’s concentration remains below the toxicity limits specified for animals, which, as established by the European Union in 2002, should remain below 300 mg/kg in terms of the total concentration in animal feedstuffs [47]. Interestingly, olive by-products had no effect on reducing SCCs when used alone but slightly reduced SCCs when used in combination with vitamin E [48]. In fact, Vit E has shown positive effects on udder health and the immune system of animals [49,50], and its combined effect with olive by-products could have a synergistic effect on mammary gland health. A positive effect on SCC was also found when grape seeds (300 g/d) were combined with 200 g of linseeds [51]. The introduction of 2% grape flour in sheep diets led to a reduction in SCCs, but no significant effect was found using 1% of the same by-product [52]. The quadratic effects of some supplements on milk SCCs suggest that the supplemented dose can have different immunostimulatory properties. Pulina et al. [53] found a reduction in SCCs in Sarda dairy sheep fed on pasture and supplemented with chestnut tannin extract. Similar results were found in a study by Castañares et al. [54], in which the supplementation of 40 g/day of chestnut and quebracho tannins to dairy sheep reduced SCCs by −14.11% and −12.03%, respectively. Condensed tannins have beneficial effects against bacteria that cause mastitis due to their ability to damage the lipid membranes of bacteria [55]. An antibacterial activity of condensed tannins against Staphylococcus aureus, Pseudomonas aeruginosa, Escherichia coli, etc., has been evidenced in vitro studies [56,57]. It remains to be elucidated which factors explain the antimicrobial activity of tannins. However, the inhibition of extracellular microbial enzymes, the disruption of peptidoglycan formation [58], the chelation of the substrates required for microbial growth [59], or efflux pump inhibition [60], concerning bacterial transport proteins that are involved in the extrusion of substrates from the cellular interior to the external environment, are all mechanisms suggested as explanations for the observed antimicrobial activities of tannins.

The beneficial effect of some tannin treatments on SCCs in sheep and, to a lesser extent, goats has been confirmed; Min et al. [61] demonstrated a reduction in the SCCs in milk from animals grazing Lespedeza cuneata (containing 15.2% of condensed tannins), a forage rich in condensed tannins. These compounds may have a bactericidal effect on the mammary gland and inhibit the proliferation of pathogens [61]. The introduction of extruded flaxseed significantly reduced (−6.89%, [62]) or did not affect [63] SCCs in milk. The supplementation of 50 g/d of coffee grounds reduced SCCs, whereas the supplementation of 100 g/d had no effect on this parameter. It has been reported that feeding coffee ground silage containing high levels of polyphenols to dairy cows with subclinical mastitis reduced milk SCCs [64]. These effects could be related to the passage of caffeine and caffeine metabolites from feed to the mammary gland and then to milk [65], which has been found to elicit various benefits for the immune system [66]. Pomegranate seed pulp had a positive effect on SCCs (−4.82%), probably due to its influence on the goat immune system, due to the polyphenols naturally present in pomegranate by-products. Similar effects have been reported among dairy cows, where the dietary supplementation of concentrated pomegranate extract (40 g/kg DM) decreased the SCCs in their milk [67]. Several phytochemicals in pomegranate such as ellagic acid and punicalagin have presented antimicrobial properties. Specifically, in vitro studies showed that punicalagin was able to counteract coagulase-positive Staphylococcus and Coagulase-negative Staphylococcus isolated from cows with mastitis due to its ability to inactivate extracellular microbial proteins [68].

In conclusion, the effect of byproducts on SCCs is quite variable and depends on the high heterogeneity of the byproducts considered and on the different doses used in the trials. Sheep and goats respond differently to the same byproduct: for example, the incorporation of cocoa by-products into a sheep diet was associated with a reduction in SCCs, whereas the same byproduct incorporated in a goat diet did not exert any effect on this parameter (Table 2). This result could be related to the animals’ different physiologies and to the different methods of trial implementation. In fact, the content of theobromine in the cocoa byproduct used in the experiment conducted by Renna et al. [69] involving goats was lower compared to that detected in the byproduct used by Carta et al. [42] in an experiment involving sheep. For this reason, the doses, the methodologies of the trials, and the methods of feeding could affect the results and have different impacts on the reduction in SCCs.

**Table 2 vetsci-10-00454-t002:** Effect of by-products’ inclusion in the diets of sheep and goats on milk somatic cell counts. Data are reported as the proportional difference between the treatment group, at the respective level of inclusion, and the control group (bold differences are considered significant, with *p* < 0.05).

Species	Dietary Treatment	g/kg of DMI	Log SCC	Milk Yield	Lactose	References
Dairy sheep	Cocoa husk 50	24	**−8.73%**	−1.52%	0.83%	[42]
	Cocoa husk 100	48	**−12.66%**	0.76%	−3.53%	
	Tomato pomace	54	8.96%	−2.21%	0.63%	[70]
	Grape marc	52	4.98%	**16.48%**	0.84%	
	Exhausted myrtle berries	41	3.98%	**−13.93%**	**−1.88%**	
	Olive cake	64	2.31%	**18.95%**	23.83%	[48]
	Olive cake + E	64 olive + 0.13 vit E	**−2.31%**	9.55%	23.83%	
	Pomegranate pulp	70	26.47%	2.41%	−1.86%	[71]
	1% of grape residue flour *	3.8	−10.20%	8.61%	0.52%	[52]
	2% of grape residue flour *	7.6	**−18.01%**	9.27%	3.47%	
	Grape seed	118	0.41%	2.01%	-	[51]
	Linseed	104	0.00%	8.72%	-	
	Mix grape and linseed	218	−11.52%	12.75%	-	
	Extruded linseed *	300 **	−0.19%	1.25%	2.08%	[72]
	Hydrolyzable tannins	60 **	**−11.88%**	0.38%	1.08%	[53]
	Hydrolyzable tannins	120 **	**−9.57%**	−0.61%	**1.95%**	
	Cocoa bean shell	45	−9.50%	-		[43]
	Pistachio shells *	50	7.50%	2.14%	0.68%	[73]
	Pomegranate hulls *	50	−14.64%	4.95%	2.04%	
	Olive pulp*	50	−1.07%	−15.64%	−8.84%	
	Olive pomace 2 phases	52.2	−19.40%	-	-	[74]
	Olive pomace 3 phases	39.6	7.38%	-	-	
	Chestnut tannins extract	66 **	−14.11%	-	-	[54]
	Quebracho tannins extract	66 **	−12.03%	-	-	
Dairy goat	Rosmary 10%	100	−2.18%	-	−2.05%	[75]
	Rosmary 20%	200	0.73%	-	**−6.97%**	
	Extruded linseed	74	2.36%	15.61%	-	[76]
	Extruded lineseed *	100 **	**−6.89%**	7.53%	12.86%	[62]
	Extruded linseed	90 **	−3.05%	1.83%	2.34%	[63]
	Pumpkin seed cake	160 **	−0.68%	14.68%	0.70%	
	Spent coffee ground 50	13.5	**−7.52%**	−2.42%	−1.13%	[77]
	Spent coffee ground 100	25.6	8.65%	3.64%	−2.72%	
	Cocoa bean shell *	92.6	2.21%	−6.08%	**−3.94%**	[69]
	Pomgranate seed pulp *	120 **	**−4.82%**	2.22%	1.68%	[78]
	Sericea lespedeza (tannins)*	-	**−19.51%**	-	-	[61]
	Artichoke silage 25%	250	2.51%	−0.47%	−0.47%	[79]
	Artichoke silage 40%	400	0.18%	2.12%	2.12%	
	Artichoke silage 60%	600	0.36%	0.24%	0.24%	

* Data originally expressed as SCCs were log-transformed by the authors. ** g/kg of concentrate. Bold values indicate significant differences (*p* < 0.05) compared to the control group as reported in the original paper.

### 3.3. Effects of Essential Oils and Vegetable and Marine Oils

The supplementation of vegetable and marine oils, as well as essential oils (EOs), in the diets of small ruminants has generally had positive effects on mammary gland health due to the oils’ potential antimicrobial properties. Some of the fatty acids present in these oils play important roles in anti-inflammatory processes, thereby improving the immunity status of udder. Sheep fed different vegetable oils presented a decrease in the SCCs in their milk [80], with the highest reduction found with flaxseed oil (−12.73%), followed by rice bran oil (−8.55%). The beneficial effect of flaxseed oil on the immune system has been well documented among cattle [81,82]: the oil increased IgG production via the improvement of immune function. The intake of n-3 PUFA could also reduce the production of TNF-α, an important mediator of the inflammatory response [82,83]. A positive effect on mammary gland health was presented by an EO complex containing thymol, eugenol, vanillin, guaiacol, and limonene; the complex significantly reduced milk SCCs in sheep [84]. A reduction in milk SCC was also observed in Lacaune sheep supplemented with 80 mg/day of curcumin, which had anti-inflammatory and antioxidant effects on the animals [85].

In dairy goats, the supplementation of coriander essential oil at a dosage of 0.95 g/kg of DM showed the greatest effect on SCC (−19.25%) [86]; this oil is rich in bioactive compounds such as linalool and geranyl acetate, which have been reported to have antioxidant, anti-inflammatory, and antimicrobial properties [87,88]. Marine algae supplementation (5.3 g/kg of DM) also reduced, although to a lesser extent, milk SCCs (−6.81%) [89]. These results demonstrated an increase in anti-inflammatory processes, probably due to n-3 PUFA in the diet, which improved mammary gland health.

The results highlighted an effective reduction in SCCs using oils, even if the extent of the reduction depends on the type of oil used and the dosage. Moreover, the differences in the components of the diet, the feeding system used, or the stage of lactation of the animals could lead to different results among sheep and goats.

**Table 3 vetsci-10-00454-t003:** Effects of essential oils inclusion in the diet of sheep and goats on milk somatic cell counts. Data are reported as the proportional difference between the treatment group, at the respective level of inclusion, and the control group (bold differences are considered significant, with *p* < 0.05).

Species	Dietary Treatment	g/kg of DM	SCC	Milk Yield	Lactose	References
Dairy sheep	Canola oil *	50 **	−2.27%	**8.47%**	0.00%	[80]
	Rice bran oil *	50 **	**−8.55%**	**8.88%**	0.00%	
	Flaxseed oil *	50 **	**−12.73%**	1.03%	−2.04%	
	Safflower oil *	50 **	−0.80%	**16.12%**	−2.04%	
	Rumen protected marine oil *	50 **	**−6.33%**	**29.75%**	−2.04%	
	Linseed oil *	20	−13.54%	-	3.85%	[90]
	Rumen protected linseed oil	6	3.19%	−7.31%	−2.44%	[91]
	Essential orange oil *	0.06	−6.22%	11.17%	−3.82%	[92]
		0.122	−0.18%	13.28%	−3.41%	
		0.200	−3.94%	4.02%	−0.80%	
	Essential oil *	0.05 **	**−3.08%**	9.56%	-	[84]
		0.1 **	**−7.13%**	**27.94%**	-	
		0.15 **	**−11.57%**	**50.74%**	-	
Dairy goat	Microalgae	5.3	**−6.81%**	4.00%	−0.22%	[89]
	Soybean oil *	26.4	−5.90%	−3.54%	1.10%	[93]
	Soybean oil+ tuna oil *	15.6 Soybean oil + 10.4 tuna oil	−0.51%	−22.87%	2.19%	
	Soybean oil+ tuna oil+ grape tannins *	15.7 Soybean oil + 10.6 tuna oil + 8.4 grape tannins	0.47%	−15.58%	−0.44%	
	Fish oil	32.2 **	−0.68%	11.90%	0.88%	[94]
	Linseed oil	100 **	−0.51%	**29.63%**	0.66%	
	L-Coriander oil *	0.95	**−19.25%**	**8.48%**	1.12%	[86]
	H-Coriander oil *	1.9	**−10.07%**	**14.01%**	1.97%	
	Pomegranade Seed oil	25	**−4.07%**	6.16%	−1.85%	[95]
	Linseed oil	25	−0.45%	5.46%	−1.44%	
	Fish oil	11	−6.99%	−10.31%	−0.42%	[96]
	Canola oil *	30	−0.70%	-	1.72%	[97]
	Sunflower oil *	30	−3.39%	-	5.41%	
	Soybean oil *	30	−1.81%	-	−0.74%	

* Data originally expressed as SCC were log-transformed by the authors. ** g/kg of concentrate. Bold values indicate significant differences (*p* < 0.05) compared to the control group as reported in the original paper.

### 3.4. Effects of Pasture and Forage to Concentrate Ratio on Milk SCCs

Experiments reporting the effects of grazing on milk SCCs are limited (Table 4). Only one experiment reported that grazing sheep had a lower milk SCC [98]. However, this difference cannot be attributed to grazing alone, as numerous confounding factors may have influenced the result. In an indoor system, animals are generally protected from adverse weather conditions, and the administration of feed is better supervised. However, the space in which animals are allocated is often insufficient, and the cleanliness and sanitary conditions are often poor; these conditions can compromise the health of the animals, leading to an increase in the SC in milk. The results presented by Casamassima et al. [98] evidenced that animals reared indoors and outdoors had different behaviors, but no differences were found regarding neutrophil and lymphocyte counts and immune response.

Large amounts of concentrate fed to high-yielding animals in one ration may increase the risk of subacute ruminal acidosis, and low-frequency concentrate feeding may reduce the potency of the immune response of the animals. For example, an increase in the plasma concentration of beta-hydroxybutyrate (BHB), an indicator of subclinical ketosis, was found to correlate with an increase in the incidence of mastitis and a decrease in the bactericidal activity of polymorphonuclear neutrophils (PMN) against mammary pathogens [11,14].

A significant, although quantitatively limited, increase in the SCC in goat milk was precipitated by a high level of concentrates in the goat diet [79], probably due to rumen subacidosis, which may impair immune function. In fact, Giger-Reverdin et al. [99] demonstrated a decrease in the fat/protein ratio of goats fed a high concentrate diet that suggested subacidosis. A relationship between rumen function and SCCs was documented by Zhong et al. [100]. They showed a reduction in volatile fatty acid concentrations in the rumens of animals with high SCCs compared to animals with low SCCs. Instead, bacterial diversity was found between the animals with different SCCs, suggesting a strong relationship between rumen fermentation and udder health.

There are few studies concerning the effect of pastures and animals’ management on SCCs. Thus, it is difficult to interpret the effects found in the studies considered in this review because there is not enough evidence to compare the results.

**Table 4 vetsci-10-00454-t004:** Effect of pasture and the forage-to-concentrate ratio in the diets of sheep and goats on milk somatic cell counts. Data are reported as the proportional difference between the treatment group, at the respective level of inclusion, and the control group (bold differences are considered significant, with *p* < 0.05).

Species	Treatment	SCC	Milk Yield	Lactose	References
Dairy sheep	Indoor *				[98]
	Outdoor *	**−10.99%**	-		
	Intensive				[101]
	Semi-intensive	0.52%	-	−1.70%	
	Concentrate-based feeding				[102]
	Artificial-pasture-based grazing	−0.49%	7.60%	0.45%	
	Feed restricted *	−15.46%	**−21.98%**	−1.92%	[36]
	High feed *	−10.80%	0.52%	−0.38%	
Dairy goat	Low concentrate (30%)				[99]
	High concentrate (60%)	**3.52%**	**23.92%**	−1.17%	

* Data originally expressed as SCC were log-transformed by the authors. Bold values indicate significant differences (*p* < 0.05) compared to the control group as reported in the original paper.

### 3.5. Combined Effect on Production Level, Milk SCCs, and Lactose Concentrations

The combined effect of the use of supplements, oils, and agro-food by-products on CSSs, production, and lactose concentrations (which have been reported in Table 1, Table 2, Table 3 and Table 4 for the experimental trials, whenever these data were available) shows few significant relationships between the three parameters analyzed. This evaluation is important because a reduction in SCC should be accompanied by an increase in lactose concentration, as a sign of improved alveolar epithelial integrity, and sometimes by an increase in milk production because of the decreased inflammatory status of the udder. In fact, the disruption of mammary epithelium integrity and the tight junction opening can be assessed using plasma lactose concentrations and the Na+ content in milk [103]. These relationships were observed for hydrolysable tannins (−9.57% SCC and +1.95% lactose, Pulina et al. [53]), rice bran (−8.55% SCC and +8.88% milk production) [80], and both doses of essential oils [84] in dairy sheep.

In goats, this combined effect only occurred with Se + Vit E (−31.79% SCC and +7.14% milk yield,) [35], high and low doses of coriander oil (less SCC and more milk) [86], and a high amount of concentrate in the rations (+3.52% SCC and +23.92% milk yield) [99].

The fact that, in many trials, the significant reduction in SCCs was not accompanied by a variation in the production and content of lactose could be explained by a direct effect of the dietary factor tested on pro-inflammatory factors, thus allowing for a lower recruitment of cellular elements towards the udder, without, however, affecting the integrity of the secretory tissues or pushing them towards higher milk production, which is an aspect that is also linked to the lactation phase and the ability of the diet tested to effectively replace that of the control group.

## 4. Conclusions

In conclusion, the dietary factors considered in this review, which are capable of reducing the SCCs in milk from dairy ewes and goats, were effective in some cases but not overall in the experimental trials considered. Mineral supplementation led to an overall reduction in SCCs, which was further enhanced by the minerals’ combined effect with vitamins. The effects of by-products are extremely variable and depend on the type of by-product used in an experiment, the amount and the types of polyphenols employed, and the doses of the by-product supplemented in the animal feed. Marine and vegetable oils showed interesting results concerning the reduction in SCCs in both sheep and goat milk. The influence of condensed tannins or essential oils on SCCs should stimulate further investigation into the antimicrobial efficacy of certain metabolites, which could lead to the development of new antimicrobials. Experiments on grazing are limited and, therefore, cannot be considered conclusive in terms of results regarding SCCs. The low correlation found between SCC, milk production, and lactose in the experiments reported could be interpreted as an effect of the diets on one of the parameters but not on the others. In general, the non-significant variations of this parameter in the experimental groups compared to the control groups were in fact greater than the significant ones, demonstrating that the immunity of the mammary gland is a complex phenomenon that cannot be resolved using a single experimental factor. One of the main confounders of the results is the presence of inflammatory phenomena of a subclinical nature in the udder, which influenced SCCs without providing the analytical feedback required in these experiments. However, the authors of this review are aware that targeted dietary supplementation with minerals, vitamins, and low doses of anti-inflammatory substances may contribute to the control of SCCs, directly contribute to the quality of the derived dairy products, and indirectly contribute to the improvement of animal welfare. Especially with the emergence of antimicrobial resistance, the use of dietary supplements as natural alternatives for disease prevention has become increasingly important.

In terms of practical implications, this review evidenced the attention that must be paid to the supplementation of minerals and vitamins in rations in order to provide adequate nutrition and support the immune system with regard to production or health considerations, without forgetting to mention the supplementation of compounds with anti-inflammatory properties, which can complement good nutrition and reduce acute local phenomena in the udder, the latter of which can be a precursor of subclinical mastitis. However, at the level of the individual farmer, the most detailed knowledge possible of the micronutrient composition of a ration is essential for the correct dosage of any supplement. This is particularly complicated to obtain with respect to small ruminants, for which most of the diet is provided by grazing wild grasses, the composition of which is highly variable in space and time. To ensure rational supplementation, limit SC in milk, and protect animal health and welfare, historical series of analyses of different types of forage and the soils in which they grow can provide useful information.

## Data Availability

Data are available upon request.

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
