# Peer review of "Feeding and Nutritional Factors That Affect Somatic Cell Counts in Milk of Sheep and Goats"

_vetsci, 2023, doi:10.3390/vetsci10070454_

Round 1

Reviewer 1 Report

Dear authors,

I have carefully reviewed your paper titled "Effects of Feeding Strategies on Somatic Cell Count in Milk of Sheep and Goats" and appreciate the valuable insights you have provided on reducing somatic cell count (SCC) in dairy sheep and goats. However, I believe that the paper requires major revisions before it can be considered for publication.

Firstly, the introduction lacks a clear and concise overview of the research problem and its significance. It is important to provide a strong rationale for studying SCC reduction in milk and highlight the potential implications for dairy production and animal health.

Secondly, the methodology section needs to be expanded to provide a detailed description of the feeding strategies employed in the studies included in your review. This should include information on the specific mineral, vitamin, marine oil, and vegetable essential oil supplements used, as well as the dosages, frequencies, and durations of supplementation. Providing this level of detail will enhance the reproducibility and applicability of your findings.

Thirdly, the results section should present a comprehensive analysis and synthesis of the included studies. It is important to provide a clear summary of the effects of the different feeding strategies on SCC in sheep and goats, highlighting any variations in response between the two species. Additionally, if any statistical analyses were conducted, the results should be reported to support your conclusions.

Furthermore, the discussion section should go beyond a general overview of the findings and provide a critical analysis of the results in relation to existing literature. Discuss the potential mechanisms underlying the observed effects of the different dietary factors on SCC and explore any inconsistencies or limitations in the studies included in your review. Additionally, consider discussing the practical implications of your findings for farmers and stakeholders in the dairy industry.

Lastly, I recommend including a conclusion section that summarizes the key findings of your review and their implications. Clearly state the practical recommendations based on your results and highlight any remaining gaps in knowledge that warrant further research.

Overall, I believe that addressing these major revisions will significantly strengthen your paper and increase its scientific rigor and relevance. I encourage you to carefully consider these suggestions and revise your manuscript accordingly.

Thank you for your valuable contribution to the field of dairy science.

Specific comments:

the affiliation must be in English

I would like to bring to your attention a recent publication that appears to be relevant to your study but was not included in your review.

The paper titled "Effect of Dietary Organic Acids and Botanicals on Metabolic Status and Milk Parameters in Mid-Late Lactating Goats" by Giorgino et al. (2023) investigates the impact of dietary organic acids and botanicals on the metabolic status and milk parameters of mid-late lactating goats. This study might provide valuable insights into the effects of specific dietary components on milk quality and could complement the findings of your review.

I would highly recommend considering this recent publication and incorporating its findings into your study. It would strengthen the comprehensiveness of your review and provide a more up-to-date perspective on the topic.

Author Response

Dear Editor

Dear Reviewers

We would like to thank the Editor and the two anonymous reviewers for their useful comments and suggestions that allowed us to improve our manuscript. The paper was changed according to the points raised by the reviewers and all changes were highlighted in yellow in the revised version.

REVIWER 1

Dear authors,

I have carefully reviewed your paper titled "Effects of Feeding Strategies on Somatic Cell Count in Milk of Sheep and Goats" and appreciate the valuable insights you have provided on reducing somatic cell count (SCC) in dairy sheep and goats. However, I believe that the paper requires major revisions before it can be considered for publication.

Firstly, the introduction lacks a clear and concise overview of the research problem and its significance. It is important to provide a strong rationale for studying SCC reduction in milk and highlight the potential implications for dairy production and animal health.

AU: Thank you for the comments. The introduction has been revised following reviewer suggestions in order to better state the objective of the study.

Secondly, the methodology section needs to be expanded to provide a detailed description of the feeding strategies employed in the studies included in your review. This should include information on the specific mineral, vitamin, marine oil, and vegetable essential oil supplements used, as well as the dosages, frequencies, and durations of supplementation. Providing this level of detail will enhance the reproducibility and applicability of your findings.

AU: Accepted. Details on type of mineral or vitamin and the dosage have been added to all tables. However, the objective of this review is to calculate the extent of variation due to a specific treatment in comparison with control group. This extent of variation quantifies the effects (in positive or negative) of a treatment independently of all factors of variations. The objective of this type of review is to synthetize and summarize the effects of similar feeding treatment. If the reader wants to have details about the effect of a specific supplement or diet, he must refer to the original work.

Thirdly, the results section should present a comprehensive analysis and synthesis of the included studies. It is important to provide a clear summary of the effects of the different feeding strategies on SCC in sheep and goats, highlighting any variations in response between the two species. Additionally, if any statistical analyses were conducted, the results should be reported to support your conclusions.

AU: Accepted. We added a section at the beginning of the result’s paragraph with a brief summary of the analysis.

The statistical differences on milk SCC due to a specific treatment is that reported by the authors. This is because is calculated on the data and the appropriate statistical analysis. This has been already stated in Material and methods section (see lines 288-290)

The differences that were statistically significant (P<0.05) are evidenced in tables in bold to be of immediate detection by the reader. A note has been added to each table.

Moreover, as request we added some to highlight variations in response between the two species. However, we prefer not to speculate on the different response between sheep and goats because of the obvious differences between species both in somatic cell content, differential cell count, and ffeding behaviour and environmental condition. It is better to look at the numbers in the table.

Furthermore, the discussion section should go beyond a general overview of the findings and provide a critical analysis of the results in relation to existing literature. Discuss the potential mechanisms underlying the observed effects of the different dietary factors on SCC and explore any inconsistencies or limitations in the studies included in your review. Additionally, consider discussing the practical implications of your findings for farmers and stakeholders in the dairy industry.

AU: Accepted. Each section of the discussion has been integrated as requested

Lastly, I recommend including a conclusion section that summarizes the key findings of your review and their implications. Clearly state the practical recommendations based on your results and highlight any remaining gaps in knowledge that warrant further research.

AU: Accepted. The Conclusion section has been integrated as suggested.

Overall, I believe that addressing these major revisions will significantly strengthen your paper and increase its scientific rigor and relevance. I encourage you to carefully consider these suggestions and revise your manuscript accordingly.

Thank you for your valuable contribution to the field of dairy science.

Specific comments:

the affiliation must be in English

I would like to bring to your attention a recent publication that appears to be relevant to your study but was not included in your review.

The paper titled "Effect of Dietary Organic Acids and Botanicals on Metabolic Status and Milk Parameters in Mid-Late Lactating Goats" by Giorgino et al. (2023) investigates the impact of dietary organic acids and botanicals on the metabolic status and milk parameters of mid-late lactating goats. This study might provide valuable insights into the effects of specific dietary components on milk quality and could complement the findings of your review.

I would highly recommend considering this recent publication and incorporating its findings into your study. It would strengthen the comprehensiveness of your review and provide a more up-to-date perspective on the topic.

Au: Accepted, we added this reference in the review.

Reviewer 2 Report

Manuscript details:

Journal: Veterinary Sciences

Manuscript ID: vetsci-2440368

Type of manuscript: Review

Title: Feeding and Nutritional Factors that affect Somatic Cell Count in milk

of sheep and goat

Authors: ANNA NUDDA *, Silvia Carta, Gianni Battacone, Giuseppe Pulina

The authors aimed to highlight the effects of feeding strategies using some mineral, vitamin, marine oil and vegetable essential oil supplements, and some agri-food by-products, on the reduction of SCC in milk of sheep and goats. Though the work is intriguing, the writers had to make changes to the document after finding a number of flaws that needed to be taken into account. I have provided suggestions below.

Abstract:

L28-32: The authors should provide the objective of this study.

L43-46: Conclusions should be based on experimental results, not on the subject's own opinion. because it will cause a feeling of being biased.

 Introduction:

L29-38: Please mention references in this sentence.

L29-38: The first paragraph seems to explain the origins, problems, and mechanisms that occur within sheep and goat cells, which is extremely interesting However, it should be written more concisely and briefly explained. Doesn't need to be explained in detail and may be written to explain the content.

L57-68: The relationship between feeding and nutritional factors that affect somatic cell count should be described more interestingly. Maybe give some examples of experiments, and what kind of result they produce.

L68: Why use specific minerals, vitamins, marine oils, and plant essential oil supplements, as well as certain agri-food by-products, in feeding lactating sheep and goats to reduce SCC in their milk? The author should initially explain the benefits of them in the introduction part.

Results and discussion

L90: The author should write a brief summary of each topic in the Results and Discussion section.

L110-117: Why using vitamin A and Zn can decrease SCC? Please explain more mechanisms.

L158: Please state the amount of theobromine found in this study.

L172: Why can tannins reduce SCC? please explain

L176: Please specify the level of condensed tannins found in this study that affects SCC.

L182: Please specify the level of polyphenols found in this study that affect SCC.

L188: Please specify the level of pomegranate extract that affects SCC.

L189-191: Why punicalagin was able to counteract coagulase-positive Staphylococcus and coagulase-negative Staphylococcus, isolated from cows with mastitis, please explain.

L210: How is IgG production related to the reduction of SCC?

L217: Please specify the level of coriander essential oil found in this study that affects SCC.

L220: Please specify the level of marine algae found in this study that affects SCC.

Moderate editing of English language required

Author Response

REVIWER 2

The authors aimed to highlight the effects of feeding strategies using some mineral, vitamin, marine oil and vegetable essential oil supplements, and some agri-food by-products, on the reduction of SCC in milk of sheep and goats. Though the work is intriguing, the writers had to make changes to the document after finding a number of flaws that needed to be taken into account. I have provided suggestions below.

Abstract:

L28-32: The authors should provide the objective of this study.

AU: The objectives are clearly stated at the beginning of the summary

L43-46: Conclusions should be based on experimental results, not on the subject's own opinion. because it will cause a feeling of being biased.

 AU: Accepted and added at the end of the summary

 Introduction:

L29-38: Please mention references in this sentence.

AU: Accepted, we added some references.

L29-38: The first paragraph seems to explain the origins, problems, and mechanisms that occur within sheep and goat cells, which is extremely interesting However, it should be written more concisely and briefly explained. Doesn't need to be explained in detail and may be written to explain the content.

AU: Thank you for the comments. We shorted the introduction, and we explain better the implication of SCC on dairy industry, accordingly to the suggestions of Reviewers 1 and 2.

L57-68: The relationship between feeding and nutritional factors that affect somatic cell count should be described more interestingly. Maybe give some examples of experiments, and what kind of result they produce.

AU: Thank you for your suggestion. In fact, to deepen this theme and not load this paper, we have added a reference to one of our previous recent works in which what is required is set out (Nudda A., Atzori A.S., Correddu S., Battacone G., Lunesu M.F., Cannas A., Pulina G. – EFFECTS OF NUTRITION ON MAIN COMPONENTS OF SHEEP MILK.  Small Ruminant Research, 2020. 184 (march): 106015  https://doi.org/10.1016/j.smallrumres.2019.11.001)

L68: Why use specific minerals, vitamins, marine oils, and plant essential oil supplements, as well as certain agri-food by-products, in feeding lactating sheep and goats to reduce SCC in their milk? The author should initially explain the benefits of them in the introduction part.

 AU: we added this explanation in the introduction.

Results and discussion

L90: The author should write a brief summary of each topic in the Results and Discussion section.

 AU: Thank you for the suggestion. We added a brief summary at the beginning of the Results and discussion paragraph.

 L110-117: Why using vitamin A and Zn can decrease SCC? Please explain more mechanisms.  

AU: Accepted. The mechanisms have been reported.

L158: Please state the amount of theobromine found in this study.

 AU: Accepted

L172: Why can tannins reduce SCC? please explain

 AU: Accepted

L176: Please specify the level of condensed tannins found in this study that affects SCC.

 AU: Accepted. The doses are reported within the tables

L182: Please specify the level of polyphenols found in this study that affect SCC.

 AU: The authors did not report this information in the paper.

L188: Please specify the level of pomegranate extract that affects SCC.

 AU: Accepted. The doses are reported within the tables

L189-191: Why punicalagin was able to counteract coagulase-positive Staphylococcus and coagulase-negative Staphylococcus, isolated from cows with mastitis, please explain.

 AU: Accepted and amended

L210: How is IgG production related to the reduction of SCC?

 AU: Accepted.

L217: Please specify the level of coriander essential oil found in this study that affects SCC.

 AU: Accepted. The doses are reported within the tables

L220: Please specify the level of marine algae found in this study that affects SCC.

 AU: Accepted. The doses are reported within the tables

Round 2

Reviewer 1 Report

The paper has improved a lot

Reviewer 2 Report

All comments​ were addressed, and recommend​ to​ accept​ for​ publication.